# Characterization of Crystal Microstructure Based on Small Angle X-ray Scattering (SAXS) Technique

**DOI:** 10.3390/molecules25030443

**Published:** 2020-01-21

**Authors:** Hongfan Wang, Jinjiang Xu, Shanhu Sun, Yanru Liu, Chunhua Zhu, Jie Li, Jie Sun, Shumin Wang, Haobin Zhang

**Affiliations:** 1School of Materials Science and Engineering, Southwest University of Science and Technology, Mianyang 621010, China; whf00300@163.com (H.W.); lyr0903211@163.com (Y.L.); shu_minwang@163.com (S.W.); 2China Academy of Engineering Physics, Institute of Chemical Materials, Mianyang 621900, China; xujinjiang@caep.cn (J.X.); shanhusun@126.com (S.S.); chzhu@caep.cn (C.Z.); li_jie@caep.cn (J.L.)

**Keywords:** SAXS, fitting, crystal powder, matching solution

## Abstract

Small-angle X-ray scattering (SAXS) is an effective method to obtain microstructural information of materials. However, due to the influence of crystal surface effects, SAXS has a deviation in the characterization of the crystal microstructure. In order to solve the influence of crystal surface effect on the internal defect signal, the microstructure of Octahydro-1,3,5,7-tetranitro-1,3,5,7-tetrazocine (HMX) crystal was characterized by soaking the sample in the matching solution. We found that the absolute scattering intensity, specific surface and volume fraction of the sample in the matching solution are significantly lower than the initial sample, which solves the influence of the crystal surface effect on the test results. Comparing the scattering results of the samples in different electron density matching solutions, it was found that the best result was obtained when using GPL-107 perfluoropolyether (PFPE) matching solution and the same law was obtained by controlling the experiment with 2,4,6,8,10,12-hexanitrohexaazaisowurtzitane (CL-20) crystal. The fitting density was calculated according to the theoretical density and void volume fraction of the sample, and the calculated results are close to the test results of Particle Density Distribution Analyzer (PDDA). Based on this paper, we provide a method to obtain the correct information of crystal microstructure.

## 1. Introduction

With the continuous requirements of industry and academia, many advanced materials with excellent properties have been developed and applied in production practice gradually [1,2,3]. It is well known that the ingenious chemical composition and structure determines their incredible application potential. However, various inevitable defects (i.e., point defects, line defects, surface defects) generated in industrial process usually decrease their practical performances such as mechanical properties, resistivity, corrosion resistance, solid-phase transformation, and other physical and chemical properties. Obviously, the density of microscopic defects is becoming one of the most important parameters to evaluate the quality of materials. Accurate microstructure information of materials is significant for evaluating their comprehensive properties and engineering applications.

Traditionally, scanning electron microscope (SEM) and transmission electron microscope (TEM) were used to characterize the microstructure of materials. SEM is designed for directly studying the surfaces of solid objects. It utilizes a beam of focused electrons of relatively low energy as an electron probe that is scanned in a regular manner over the specimen. It can be used to observe particles at a much higher magnification and resolution than can be achieved with a light microscope because the wavelength of an electron is much shorter than that of a photon. Its resolution can reach nanometer level when the conductive material is excited at high voltage. Similarly, TEM is a technique used to observe the features of very small specimens. It used an accelerated beam of electrons, which passes through a very thin specimen to enable a scientist to observe features such as structure and morphology. It also provides higher resolution images than SEM. Using TEM, scientists can be used to view specimens to the atomic level, which is less than 1 nm. Nevertheless, electrons with high energy usually damage some soft materials, and the analysis results cannot reflect their original structure. Of late years, small-angle X-ray scattering (SAXS) technology has gradually attracted people’s attention due to its advantages of good statistics, simple sample preparation and non-destructive testing. Based on the coherent scattering near the incident beam (usually 0.05^°^–5^°^), it is extremely sensitive to the scattering length density variation of several nanometers to several hundred nanometers. It is becoming the ideal candidate to supplement the technique of SEM and TEM. At present, SAXS has been widely used as an advanced means for obtaining specific information on the number and size distribution of internal defects of materials, such as polymer materials [4,5,6,7,8,9], nanocomposites [10,11,12] and energetic materials [13,14,15,16,17,18,19,20,21]. However, it is still very challenging for characterizing the crystal powder accuracy by SAXS. On the one hand, the absolute scattering intensity should be corrected. On the other hand, because the surface effect of crystal affects the scattered signal, the test results cannot reflect the real structure inside the crystal.

In the field of energetic materials, the hot spots [22,23,24] caused by microscopic defects in explosive crystals usually lead to negative effects on the properties, such as improved shock wave sensitivity [25], decreased mechanical properties [26,27] and thermal stability [28]. In this work, in order to characterize the quality of explosive crystals, SAXS technology was employed. To remove interference from the cracks and voids between particles, some match solutions with different electron densities (i.e., cyclohexane, polydimethylsiloxane and GPL-107 perfluoropolyether) was used to infiltrate the crystal powders of CL-20 and HMX. It was found that the SAXS results were accurate when the electron density of the matching solution was close to that of the sample. Further, the real scattering data of the inner structure of the explosive crystal were obtained by correcting the absolute scattering intensity of the sample and comparing the change of the specific surface and volume fraction of the voids. 

## 2. Results and Discussion

### 2.1. Research on Microstructure of HMX

In small-angle X-ray scattering technique, the measured scattering intensity of sample is only relative intensity, which is related to experimental conditions and sample size (such as thickness). Absolute intensity is the ratio of scattered light intensity to incident ray intensity, also known as unit volume differential scattering cross section, which is similar to Rayleigh ratio in light scattering. It is only related to the nature of the sample itself, and has nothing to do with the external conditions such as test instrument, exposure time and sample size. Absolute scattering intensity is used to calculate parameters related to mass density, such as relative molecular mass [29,30] and electron density difference [31]. Absolute intensity is also called unit volume differential scattering cross section. The differential scattering cross section is defined [32] as the ratio of optical intensity scattered per unit time unit solid angle to incident light intensity. Based on the above definition, we can obtain the relationship between the absolute scattering intensity and the measured relative intensity:(1)$$\frac{d\Sigma }{d\Omega }=\frac{\left(I_{s}(q)/T_{s}-I_{bg}(q)/T_{bg}\right)}{I_{0}\;d\;t\;p_{1}\;p_{2}}\frac{L^{3}{}_{p}}{L_{0}}$$
where *d*Σ/*d*Ω is the absolute scattering intensity; *I_s_* is the relative scattering intensity of the sample; *I_bg_* is the relative scattering intensity of the background; *T_s_* and *T_bg_* are the transmittance of the sample and the background, respectively; *I*_0_ is the incident ray intensity (counts s^−1^); *d* is the thickness of the sample; t is the exposure time; *P*_1_ and *P*_2_ are the sizes of the pixel in the horizontal and vertical directions; *L*_0_ is the distance from the sample to the detector; *L_P_* is the distance of the sample to a pixel. The structural parameters of the scattering system can be accurately calculated from the corrected absolute scattering intensity. Therefore, absolute scattering intensity correction is essential.

Appendix A shows the selected SAXS patterns of HMX samples in different matching solutions. Since all samples are tested under the same conditions, the range of scattering vector q (q = 4πsinθ/λ, where 2θ is the scattering angle and λ is the wavelength of the X-ray) is the same. The scattering invariant *Q* is defined as follows:(2)$$Q=\int_{0}^{\infty }\frac{d\Sigma }{d\Omega }(q)q^{2}dq$$

It can be known that the scattering invariant *Q* is consistent with the change law of the absolute scattering intensity. The calculation results of the scattering invariant *Q* in different electron density matching solution of HMX crystal are shown in Figure 1. The results showed that the absolute scattering intensity of the HMX crystal which is not infiltrated by the matching solution is highest. It proves that apart from the voids in HMX crystal itself, the crystal surface effect also affects the scattering results. From the principle of SAXS technology, it can be recognized that the scattered signal of the sample comes from the difference in electron density between the two phases. The electron density can be obtained according to:(3)$$\rho_{e}=E\times \frac{\rho_{m}}{M}\times 6.022\times 10^{23}$$
where *E* is the atomic number; *M* is the molar mass; *ρ_m_* is the mass density. Absolute scattering intensity is significantly reduced when HMX crystals are infiltrated into the matching solution. With the constant change of electron density of matching solution, the absolute scattering intensity also changed gradually. It is showed that in addition to the scattering signal of the internal voids of the crystal, the difference of electron density between the matching solution and the crystal also contributes to the scattering signal. The theoretical density of β-HMX at 20 °C is about 1.902 g/cm^3^, and the corresponding electron density is 588.170 nm^−3^. It was found that the absolute scattering intensity was the weakest when the electron density of the matched solution was 571.727 nm^−3^. It is proved that the smaller the difference of electron density between the matching solution and sample, the smaller contribution to the scattering signal. Therefore, the scattering signal of HMX crystal is relatively accurate only in the PFPE matching solution.

In addition, we also studied the volume fraction and specific surface of the voids in HMX. The volume fraction of voids can be obtained according to:(4)$$Q=2\pi^{2}I_{e}{\left(\rho_{A}-\rho_{B}\right)}^{2}\varphi_{A}\varphi_{B}$$
where *φ_A_* and *φ_B_* denote the volume fraction of the sample and the voids, and *ρ_A_* and *ρ_B_* are the corresponding electron density, respectively. *I_e_* is the scattering intensity of an electron:(5)$$I_{e}=r_{e}{}^{2}\frac{1+\cos^{2}2\theta }{2}=7.90\times 10^{26}\times \frac{1+\cos^{2}2\theta }{2}$$
where *r_e_* is a classical electron radius with a size of 2.818 × 10^−13^ cm. The polarization factor is approximately 1 in the small angle region. According to Porod method, the specific surface *S*/*V* of the two-phase system can be obtained by the following equation:(6)$$\frac{S}{V}=\pi \varphi_{A}\varphi_{B}\frac{K_{P}}{Q}$$
where *K_p_* is the Porod constant.

From Equations (4)–(6), we can calculate the specific surface and the volume fraction of the voids in the crystal. The results are shown in Figure 2. When HMX crystal was not infiltrated by matched solution, the volume fraction and specific surface of the voids were the largest, which was 1.495% and 5241.280 cm^−1^, respectively. With the increase of electron density of the matched solution, the volume fraction and the specific surface of the voids decreased. The results show that the volume fraction and the specific surface of the HMX crystal voids are inaccurate due to the surface effect of the crystal without the addition of a matching solution. The closer electron density between the matching solution and the crystal can be expected to result in smaller the volume fraction and specific surface of the voids. It is indicated that the electron density difference between the matching solution and the sample has an effect on the calculated results. The volume fraction and the specific surface of the voids in the perfluoropolyether matching solution were the smallest, 0.130% and 201.580 cm^−1^, respectively. 

According to Equations (2) and (4), the calculation result of void volume fraction is related to the scattering invariant *Q*, and the scattering invariant *Q* is related to the absolute scattering intensity of the sample. Since it is difficult to find a matching solution with the same electron density as the HMX crystal, the absolute scattering intensity results obtained are often affected by the difference in electron density between the HMX crystal and the matching solution. In order to obtain the true volume fraction of the crystal voids, we studied the influence of the electron density difference between the HMX crystal and the matching solution on the scattering signal. Equations (7)–(9) is transformed from Equation (4).
(7)$$Q_{1}=2\pi^{2}I_{e}{\left(\rho_{1}-\rho_{2}\right)}^{2}\varphi_{1}\varphi_{2}=b$$
(8)$$Q_{2}=2\pi^{2}I_{e}{\left(\rho_{1}-\rho_{3}\right)}^{2}\varphi_{1}\varphi_{3}=a\Delta \rho^{2}$$
(9)$$Q=Q_{1}+Q_{2}=a\Delta \rho^{2}+b$$
where *ρ*_3_ and *φ*_3_ are the electron density and volume fraction of the matching solution; *φ*_1_ and *φ*_2_ denote the volume fraction of HMX crystals and the voids, and *ρ*_1_ and *ρ*_2_ are the corresponding electron density, respectively. Scattering invariant *Q*_1_ depends on the absolute scattering intensity of the voids inside the HMX explosive crystal. Since the voids inside the crystal depend on the nature of the sample itself, we assume that *Q*_1_ is a constant b. The scattering invariant *Q*_2_ depends on the scattered signal caused by the difference in electron density between the HMX crystal and matching solution, which we assume is *a*Δ*ρ*^2^. Therefore, we transform the obtained scattering invariant *Q* into a linear regression equation. Figure 3 shows the scattering invariant *Q* for HMX crystal varies with the square of the electron density difference:

As can be seen, linear fitting of the data has a good result. The results show that the scattering constant *Q* is proportional to the square of the electron density difference between the HMX crystal and the matching solution. The larger the electron density difference is, the greater the influence on the scattering signal of the crystal will be. The true volume fraction of the voids in HMX crystal calculated by intercept b is 0.130%. The fitting density of HMX crystal is calculated according to Equations (10)–(14):(10)$$\rho_{m,1}V_{1}=m_{1}$$
(11)$$\rho_{m,2}V_{2}=m_{2}$$
where *ρ_m_*_,1_, *V*_1_, and *m*_1_ represent the theoretical density, volume, and mass of the HMX crystal, respectively; *ρ_m_*_,2_, *V*_2_, and *m*_2_ are the density, volume, and mass of the voids, respectively. Since the voids mass and density are both zero, Equations (10) and (11) can be converted to:(12)$$\rho_{m}{{}_{,}}_{3}(V_{1}+V_{2})=\rho_{m,1}V_{1}$$
(13)$$\frac{V_{1}}{V_{1}+V_{2}}=1-\varphi_{B}$$
(14)$$\rho_{m,3}=\rho_{m,1}(1-\varphi_{B})$$
where *ρ_m_*_,3_ is the true mass density of the HMX crystal. We calculated that the fitting density of the HMX is 1.8995 g/cm^3^. The PDDA was used to test the HMX crystal and the true density was 1.8993 g/cm^3^, which was consistent with the fitting density. Appendix A shows the mass density distribution of HMX crystal. By using the void volume fraction of the HMX crystal that was not infiltrated by the matching solution, the calculated density was 1.8736 g/cm^3^, which was smaller than the true density of the sample. Obviously, the result is incorrect. The results of HMX are shown in Table 1.

### 2.2. Research on Microstructure of CL-20 Crystal with Low Quality

In order to verify the applicability of the method, we selected CL-20-1 crystal for comparative experiments. Appendix A shows the selected SAXS patterns of CL-20-1 samples in different matching solutions. Figure 4 shows the calculation results of the scattering invariant *Q* in different electron density matching solution of CL-20-1 crystal. It can be seen that the absolute scattering intensity of CL-20-1 crystal in the matching solution is significantly reduced. The theoretical density of ε-CL-20 at 20 °C is about 2.040 g/cm^3^, and the corresponding electron density is 622.658 nm^−3^. We found that the closer the electron density of the matching solution and the CL-20-1 crystal is, the lower the absolute scattering intensity will be. The calculation results of specific surface and the volume fraction of CL-20-1 crystal voids are shown in Figure 5. It can be found that the specific surface and volume fraction of the voids were the maximum when the CL-20-1 crystal was not infiltrated by the matched solution, which was 1.903% and 9840.020 cm^−1^, respectively. The closer electron density of the matching solution and CL-20-1 crystal can result in the smaller volume fraction and the specific surface of the voids. The volume fraction and specific surface of the voids in the perfluoropolyether matching solution were the smallest, 0.356% and 3315.550 cm^−1^, respectively. It can be concluded that with the increase of the electron density of the matching solution, the specific surface and volume fraction of the void in CL-20-1 crystal decrease, which is consistent with the change law of HMX crystal. Figure 6 shows the scattering invariant *Q* for CL-20-1 crystal varies with the square of the electron density difference:

The results of CL-20-1 are shown in Table 2. As we can see, the data fits linearly with good results. The true volume fraction of the voids in the crystal calculated from the intercept b is 0.405%. The fitting density of CL-20-1 crystal is calculated as 2.0317 g/cm^3^. The PDDA was used to measure CL-20-1 crystal, and the true density was 2.0080 g/cm^3^, which was smaller than the fitting density. Appendix A shows the mass density distribution of CL-20-1. The calculated density is relatively high due to the following two reasons: ①Some large defects in the crystal exceed the detection limit of the instrument, and the relevant SAXS data cannot be obtained. ②CL-20-1 crystal has a certain density difference, the test sample itself density is not uniform. But the HMX test deviation was significantly lower than that of CL-20-1 crystal, which indicated that good quality of crystal was conducive to accurate SAXS test results.

### 2.3. Research on Microstructure of CL-20 with Different Sizes

According to the above results, we selected GPL-107 perfluoropolyether reagent as the matching solution and tested CL-20 crystals of the same quality and different particle size with SAXS technology. Appendix A shows the selected SAXS patterns of CL-20 samples with different particle sizes in different matching solutions. The GPL-107 perfluoropolyether reagent was selected as the matching solution due to its high electron density and the lower electron density contrast between itself and CL-20 crystal. This enabled us to investigate the voids in crystal using SAXS without too much interference of the scattering from the difference in electron density between the crystal and the matching solution.

Figure 7 shows the results of the void volume fraction and specific surface of CL-20 crystals with different particle sizes. Since the crystal surface effect will have a significant impact on the SAXS test results, the consequences of the crystal without infiltrating the matching solution have a lot of randomness. After infiltrating in the PFPE matching solution, the volume fraction and the specific surface of the voids in the crystals decreases with the increase of the crystal size. At the same temperature, the theoretical density of CL-20 crystals with different particle sizes is identical. It can be known from Equation (14) that the larger volume fraction of the voids can give rise to the smaller density of the crystals. Therefore, the density of the crystal should decrease as the particle size increase. Table 3 presents the results of CL-20 crystals with different particle sizes. Based on the results calculated in PFPE, we can see that the density of the CL-20-4 is the highest, and the density of the CL-20-2 crystal is the lowest, which is consistent with the regular of true density. The mass density distribution of CL-20-2, CL-20-3, and CL-20-4 are given in Appendix A, respectively. These results confirm our speculation and show that the data are correct. Because the quality of sample is better, the deviation between the calculated density and the true density is small.

## 3. Experimental

Raw CL-20 and HMX were provided by the Institute of Chemical Materials, Chinese Academy of Engineering Physics (CAEP). Cyclohexane (*ρ_e_* = 268.409 nm^−3^) was provided by Sinopharm Chemical Reagent Factory. Polydimethylsiloxane (*ρ_e_* = 325.514 nm^−3^) was purchased from Dow corning and GPL-107 perfluoropolyether (*ρ_e_* = 571.727 nm^−3^) was purchased from Banliwei information technology co. LTD. CL-20-1 is a kind of explosive crystal with many defects, which is used for methodology research. CL-20-2, CL-20-3 and CL-20-4 are crystals of 10 μm, 200 μm and 300 μm, respectively.

The true density of CL-20 and HMX crystal was taken with density gradient column floats in PDDA-SMART Particle Density Distribution Analyzer (PDDA). The density gradient solution is configured by zinc bromide aqueous solution. Figure 8 shows the principle of PDDA.

The SAXS experiments were conducted with a Xeuss system of Xenocs 2.0, France. The system is equipped with a multilayer focused Cu Kα X-ray source (GeniX^3D^ λ = 0.154 nm), generated at 50 kV and 0.6 mA. Two pairs of scatterless slits were located 1500 mm apart from each other for collimating the X-ray beam. Scattering data were recorded with the aid of a Pilatus 300 K detector (resolution: 487 × 618, pixel size = 172 µm). In order to get a larger size of information, we shifted the Beam stop up slightly. The effective range of the characteristic dimension was 10–400 nm. The sample-to-detector distance was 2490 mm. The collection time for SAXS patterns was set as 600 s after each step. The CL-20 crystal was sufficiently wetted in cyclohexane, polydimethylsiloxane, and GPL-107 perfluoropolyether matching solution, and packaged in an iron piece having a diameter of 2 mm and a height of 1mm using a Kapton Tape. A sample without any treatment was selected for comparison and the CL-20 crystal is treated in the same way as HMX crystal. The X-rays are scattered by the sample and recorded as two-dimensional data by the detector, and processed into a one-dimensional curve by the software Fit-2D. Appendix A show the 1D SAXS curves of HMX, CL-20-1, and CL-20 with different particle sizes in different matching solutions, respectively.

## 4. Conclusions

In this paper, the scattering data which can reflect the real structure inside the crystal is obtained by infiltrating the crystal into the matching solution. It was found that the influence of crystal surface effect on the test results can be effectively solved. It is worth noting that the electron density of the matching solution should be the same or similar to the electron density of the sample. By fitting the scattering invariant *Q* of CL-20 and HMX crystals in different electron density matching solutions, the volume fraction of the voids inside the crystal was obtained. The fitting density of the crystal is close to the test result of PDDA, which proves the effectiveness of the method. As for CL-20 crystals with different particle sizes, SAXS results show that scattering data with small deviation can be obtained in GPL-107 perfluoropolyether matching solution. This method is suitable for SAXS research of crystals with different types and sizes.

## Figures and Tables

**Figure 1 molecules-25-00443-f001:**
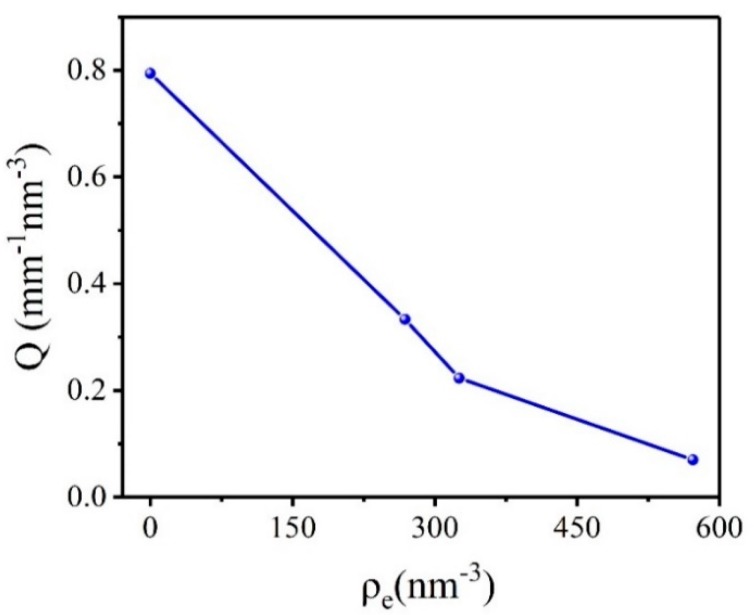
Scattering invariant *Q* of HMX crystal in different electron density matching solutions.

**Figure 2 molecules-25-00443-f002:**
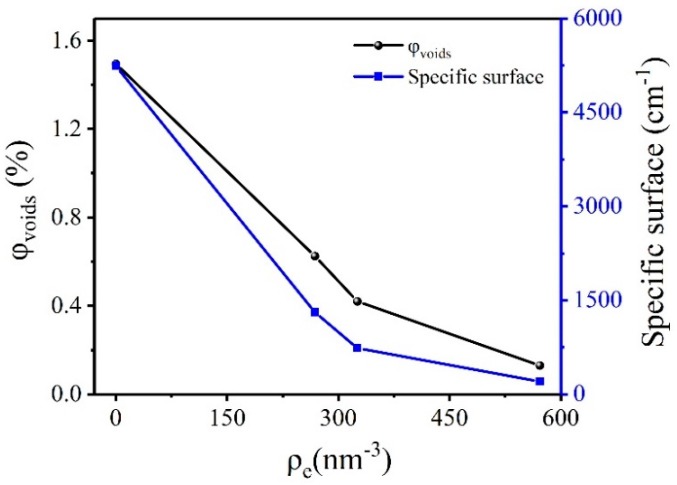
Volume fraction and specific surface of voids in HMX crystals in different electron density matching solutions.

**Figure 3 molecules-25-00443-f003:**
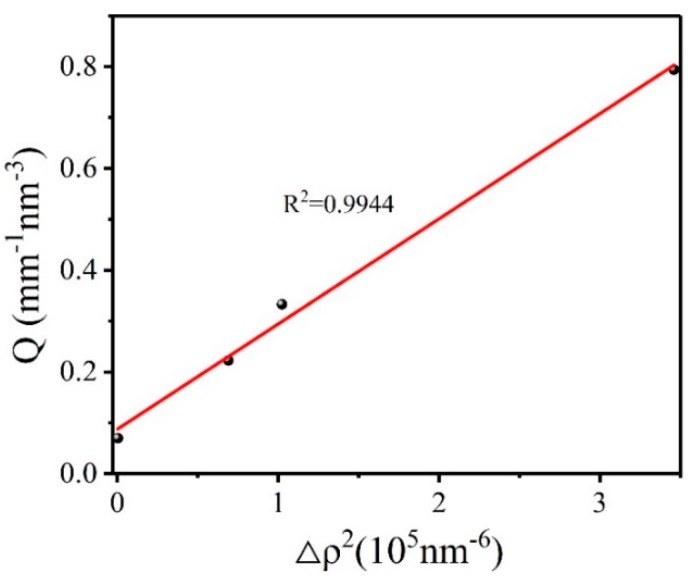
Scattering invariant *Q* for HMX crystal as a function of electron density difference.

**Figure 4 molecules-25-00443-f004:**
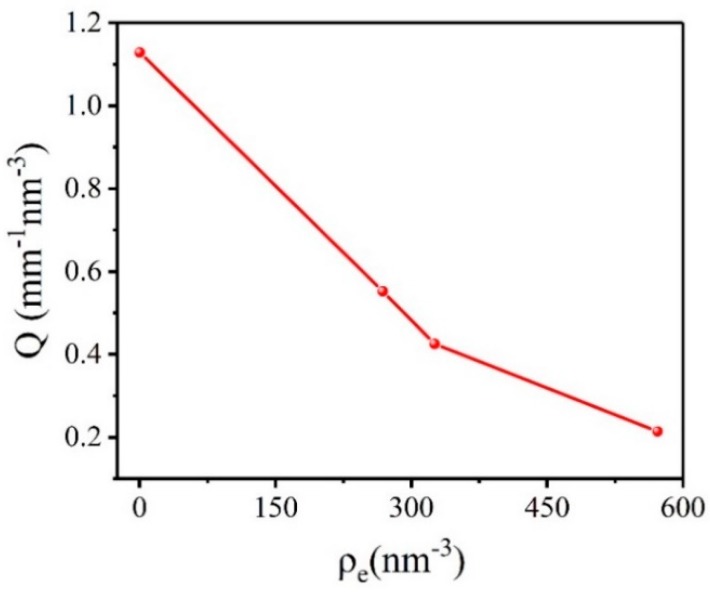
Scattering invariant *Q* of CL-20-1 crystal in different electron density matching solution.

**Figure 5 molecules-25-00443-f005:**
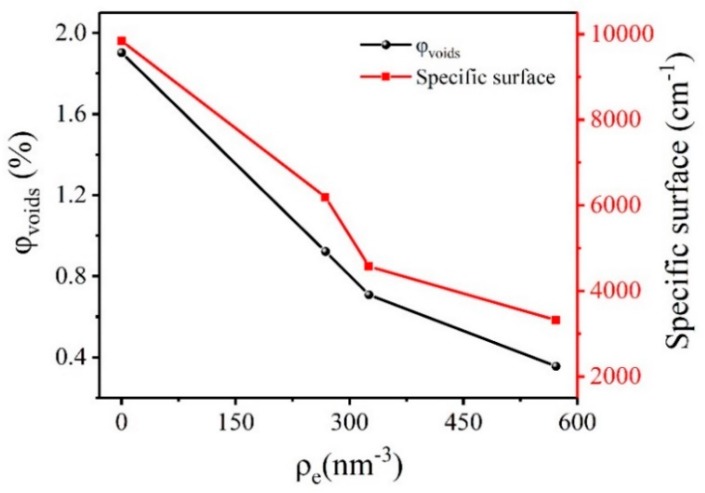
Volume fraction and specific surface of voids in CL-20-1 crystals in different electron density matching solutions.

**Figure 6 molecules-25-00443-f006:**
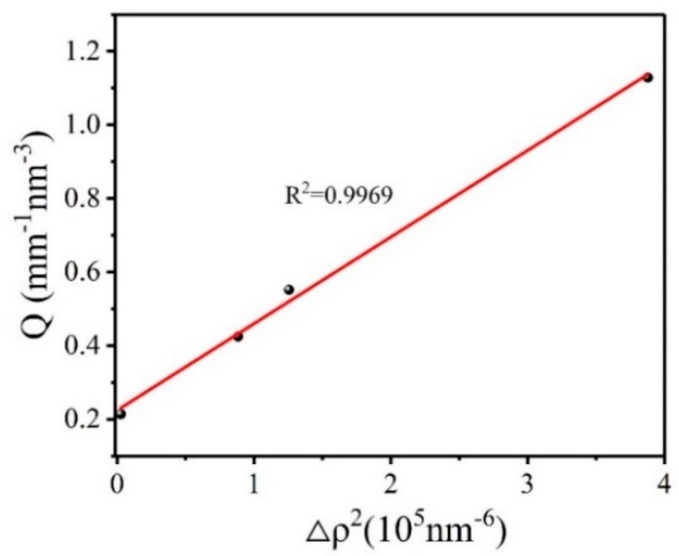
Scattering invariant *Q* for CL-20-1 crystal as a function of electron density difference.

**Figure 7 molecules-25-00443-f007:**
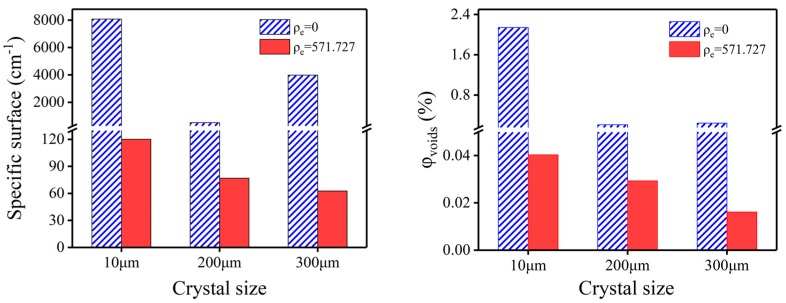
Specific surface (left) and volume fraction (right) of voids in air and perfluoropolyether matching solution of CL-20 crystals with different particle sizes.

**Figure 8 molecules-25-00443-f008:**
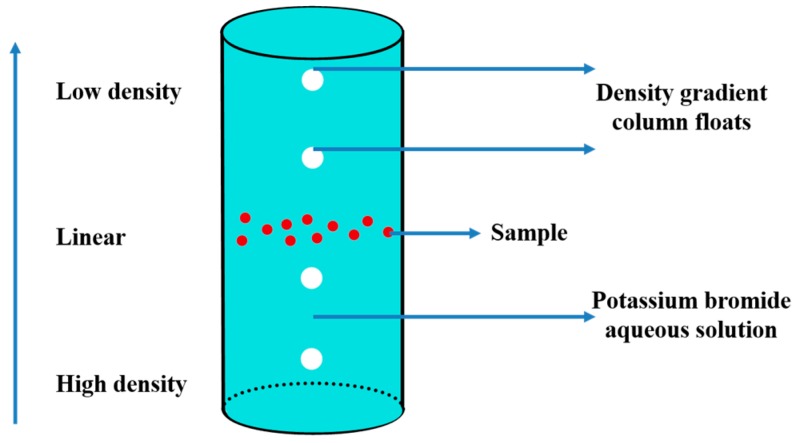
Schematic diagram of PDDA.

**Table 1 molecules-25-00443-t001:** Calculation results of parameters of 1,3,5,7-tetranitro-1,3,5,7-tetrazacyclooctane (HMX) crystal under different conditions.

Sample	Parameter	Untreat	Cyclohexane	Polydimethylsiloxane	Pfpe
	*Q* (mm^−1^ nm^−3^)	0.794	0.333	0.223	0.07
	Void volume fraction	1.495%	0.625%	0.420%	0.130%
**HMX**	Specific surface (cm^−1^)	5241.280	1308.420	736.830	201.580
	*ρ_m,c_* (g/cm^3^)	1.8736	1.8901	1.8940	1.8995
	*ρ_m,t_* (g/cm^3^)	1.8993			

*ρ_m,c_* is the mass density of the sample in the air and different matching solution; *ρ_m,t_* is the true mass density obtained by PDDA; *ρ_m,c_* and *ρ_m,t_* have the same meaning in all the tables in the article.

**Table 2 molecules-25-00443-t002:** Calculation results of parameters of CL-20-1 crystal under different conditions.

Sample	Parameter	Untreat	Cyclohexane	Polydimethylsiloxane	Pfpe
	*Q* (mm^−1^ nm^−3^)	1.129	0.552	0.425	0.214
	Void volume fraction	1.903%	0.922%	0.708%	0.356%
**CL-20-1**	Specific surface (cm^−1^)	9840.020	6190.500	4568.880	3315.550
	*ρ_m,c_* (g/cm^3^)	2.0012	2.0212	2.0256	2.0327
	*ρ_m,t_* (g/cm^3^)	2.0080			

**Table 3 molecules-25-00443-t003:** Parameter calculation results of CL-20 crystal with different particle size under different conditions.

Sample	Condition	Void Volume Fraction	Specific Surface(cm^−1^)	*ρ_m,c_* (g/cm^3^)	*ρ_m,t_* (g/cm^3^)
CL-20-2 (10 μm)	Untreat	2.140%	8077.501	1.9963	2.0368
	PFPE	0.0403%	120.213	2.0392	
CL-20-3 (200 μm)	Untreat	0.210%	512.591	2.0357	2.0371
	PFPE	0.0293%	76.953	2.0394	
CL-20-4 (300 μm)	Untreat	0.241%	3985.635	2.0351	2.0380
	PFPE	0.0161%	62.666	2.0396	

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
