# Peer review of "Characterization of Crystal Microstructure Based on Small Angle X-ray Scattering (SAXS) Technique"

_molecules, 2020, doi:10.3390/molecules25030443_

Round 1

Reviewer 1 Report

The authors tried to characterise the microscopic defects of explosive crystals. They applied Small Angle X-ray scattering. They varied matching solutions, to vary the electron contrast of crystals and the background by varying the matching solutions. In particular they carried the electronic contrast. Crystal powders of CL-20 and HMX were soaked in different matching solutions. They compared results obtained from SAXS data results derived from particle density distribution analyser.  

The authors do not provide any scattering intensity data. I did not find the supplementaries. The paper is full of typos. One can anticipate the content, but I am not sure if the authors truly meant what I anticipate. 

The paper is hard to read. It lacks a logical setup. Why did the authors chose SAXS? Voids of a crystal could be also accessed by varying the densities of the “matching” solutions and then measuring the difference in weight of the samples.

Some errors, exemplarily, but there are more.   

Line 59, 61 “Because the relative scattering intensity of the test sample can only obtain the geometric parameters of the scatterer, the parameters related to the mass density cannot be obtained.”  I disagree at least with what I understand. If the authors think of Debye’s Formular, the weighting parameters, are pretty much related to the mass densities, of the individual sites.  

Line 69, 70 “SAXS results were more well” I do not thing that this is an appropriate wording.

Line 113, 113 Equation 1 Is not readable 

Line 115 “…; Is is” typo should read, it is. 

Line  122, 123 Whereof did the authors get the equation?

Line 126, 127 There is no ρ in Equation 4

Line 135 "Kp is the Period Constant” Do the authors refer to Porod Constant?

Line 199  “… into a binary one-order equation” Is this a linear Regression?

Line 209, Equation 9 10, m1 and m2 are missing, Note the additional in the Text typos.

Line 212. 213 “Because the voids mass and density are both zero …” Why should the mass of a void be not zero?

Line 223 Table 2 is not readable. 

Line 277 “It can be known from equation (11) that the larger volume fraction of the voids can give rise to the smaller density of the crystals.”  I do not see how the authors conclude this from Eq 11. 

I do not see a point to publish this work in the present state. I do not question the validity of their experimental efforts. There are no experimental data shown. Possibly, it would be better to pack the experimental evidences into the actual work and not in the supplementaries. The latter I could not access. 

Author Response

We thank all Reviewers for their careful and favorable reviews of our manuscript. We have addressed their questions and comments and have further improved our manuscript. Supporting material is displayed at the bottom of the article for reviewers' reference. Our explicit responses to each of the Reviewers’ comments are shown below.

Point-by-Point Responses to Reviewers:

Responses to Reviewer 1:

Recommendation: I do not see a point to publish this work in the present state.

Comments:

The authors tried to characterise the microscopic defects of explosive crystals. They applied Small Angle X-ray scattering. They varied matching solutions, to vary the electron contrast of crystals and the background by varying the matching solutions. In particular they carried the electronic contrast. Crystal powders of CL-20 and HMX were soaked in different matching solutions. They compared results obtained from SAXS data results derived from particle density distribution analyser.  

Response: Thanks for the reviewer’ suggestions and we will correct these mistakes in our revised version. Following the reviewer’ suggestions, we address the detailed language problems raised by Reviewer 1 below:

The paper is hard to read. It lacks a logical setup.

Thanks for the reviewer’s comments. We corrected the absolute scattering intensity. Because the scattering constant Q is positively correlated with the absolute scattering intensity, the scattering invariant Q is consistent with the change law of the absolute scattering intensity. Based on the scattering invariant Q, we can calculate the volume fraction of the void. The calculated density of the sample was obtained from the volume fraction of the void. The calculated density is basically the same as the density obtained by the PDDA test, which proves that the method and SAXS data are correct.

2.Why did the authors chose SAXS? Voids of a crystal could be also accessed by varying the densities of the “matching” solutions and then measuring the difference in weight of the samples.

Thanks for the reviewer’s comments. Of late years, Small-angle X-ray scattering (SAXS) technology has gradually attracted people's attention due to its advantages of good statistics, simple sample preparation and non-destructive testing. Based on the coherent scattering near the incident beam (usually 0.05o ~ 5o), it is extremely sensitive to the scattering length density variation of several nanometers to several hundred nanometers. It is becoming the ideal candidate to supplement the technique of SEM and TEM. At present, SAXS has been widely used as an advanced means for obtaining specific information on the number and size distribution of internal defects of materials, such as polymer materials, nanocomposites and energetic materials.

Line 59, 61 Because the relative scattering intensity of the test sample can only obtain the geometric parameters of the scatterer, the parameters related to the mass density cannot be obtained.”I disagree at least with what I understand. If the authors think of Debye’s Formular, the weighting parameters, are pretty much related to the mass densities, of the individual sites.

Thanks for the reviewer’s comments. Because the author's work in the references is not perfect, the explanation of the professional term is not accurate. We have deleted the unnecessary sentence of “Because the relative scattering intensity of the test sample can only obtain the geometric parameters of the scatterer, the parameters related to the mass density cannot be obtained” in page 2 line 60.

Line 69, 70 “SAXS results were more well” I do not thing that this is an appropriate wording.

Thanks for the suggestion. We have changed the “more well” into “accurate” in page 2 line 71.

Line 113, Equation 1 Is not readable

Thanks for the reviewer’s comments. We have adjusted the format of the equation in page 3 line 115.

Line 115 “…; Is is” typo should read, it is.

Thanks for the reviewer’s suggestion. we have revised the “Is” into “Is” in page 3 line 116.

Line122, 123 Where of did the authors get the equation?

Thanks for the reviewer’s comments. This equation can be obtained from reference [7].

Line 126, 127 There is no ρ in Equation 4

Thanks for the reviewer’s comments. we have revised the “ρ” into “ρm” in page 4 line 133.

Line 135 "Kp is the Period Constant” Do the authors refer to Porod Constant?

Thanks for the reviewer’s comments. we have revised the “Period” into “Porod” in page 4 line 137.

Line 199“… into a binary one-order equation” Is this a linear Regression?

Thanks for the suggestion. We have changed the “binary one-order equation” into “linear regression equation” in page 6 line 200.

Line 209, Equation 9 10, mand mare missing, Note the additional in the Text typos.

Thanks for the reviewer’s suggestion. We have modified the format of the equation in page 6 line 211.

Line 212. 213 “Because the voids mass and density are both zero …” Why should the mass of a void be not zero?

Thanks for the reviewer’s comments. We think that there is only air in the void, and the void itself is massless.

Reviewer 2 Report

In the manuscript authors used different matching solutions to calculate volume fraction and specific surface of voids in the powder crystalline material HMX and CL-20-1 by means of SAXS. By treating Q1 as a constant and Q2 as a difference in electron density between a crystal and a solution authors make extrapolation of the measured points, which results in better prediction of volume fraction and specific surface of voids.  Authors also tested the size of particle on the measured parameters. The manuscript is interesting and after minor correction should be published.

Minor isues:

l.52 (usually 0.05o ~ 5o), change to upper index

l.54 "It is becoming the ideal candidate to fill the deficiencies between the resolution of SEM and TEM." SAXS is more like addition to the two techniques not a gap fill between them.

l76,77,78 "(ρe=268.409 nm-3)" correct index -3

l115,116 please use down indices as in equation

l113 correct the equation there is a shift in notation

l123 ρA and ρB are not as in equation

l136 make the chapter in bold or bigger for clarity

Equation 1,2 what is the q? is not in a descripton

Table 1 -please change the column titles to smaller size, correct indices

Figure 2,3. Please add error bars to the points or explain why you don't use them

l194 index Q1

l198 aΔρ2 index this really makes difference

l209 please correct the shift 

l223 pleas correct the size of titles in table 2. A lot of shifts

l243 indeces

l243 "It can be concluded that the regular of CL-243 20-1 crystal is consistent with that of HMX crystal." you mean the results are consistent?

l258 Make title specific
